# Measuring Impairments of Mentalization with the 15-Item Mentalization Questionnaire (MZQ) and Introducing the MZQ-6 Short Scale: Reliability, Validity and Norm Values Based on a Representative Sample of the German Population

**DOI:** 10.3390/diagnostics13010135

**Published:** 2022-12-30

**Authors:** David Riedl, Hanna Kampling, Tobias Nolte, Astrid Lampe, Manfred E. Beutel, Elmar Brähler, Johannes Kruse

**Affiliations:** 1Ludwig Boltzmann Institute for Rehabilitation Research, 1100 Vienna, Austria; 2University Hospital of Psychiatry II, Department of Psychiatry, Psychotherapy, Psychosomatics and Medical Psychology, Medical University of Innsbruck, 6020 Innsbruck, Austria; 3Department of Psychosomatic Medicine and Psychotherapy, Justus Liebig University Giessen, 35390 Giessen, Germany; 4Wellcome Department of Imaging Neuroscience, University College London, London WC1N 3AR, UK; 5Anna Freud National Centre for Children and Families, London N1 9JH, UK; 6VAMED Rehabilitation Center, 6780 Schruns, Austria; 7Department of Psychosomatic Medicine and Psychotherapy, University Medical Center, Johannes Gutenberg University of Mainz, 55122 Mainz, Germany; 8Department of Psychiatry and Psychotherapy, University of Leipzig Medical Center, 04103 Leipzig, Germany; 9Department for Psychosomatic Medicine and Psychotherapy, Medical Center of the Philipps University Marburg, 35037 Marburg, Germany

**Keywords:** mentalization, mental disorders, MZQ, norm values, reliability, validity

## Abstract

Deficits in mentalization are indicated by impaired emotional awareness and self-reflectiveness, and are associated with various mental disorders. However, there is a lack of validated research instruments. In this study, the psychometric properties of the Mentalization Questionnaire (MZQ) were evaluated in a representative German population sample with n = 2487 participants. Analyses included evaluation of the MZQs acceptance, reliability, and validity. Factorial validity was established with exploratory (EFA) and confirmatory factor analyses (CFA) after the dataset was randomly split. Dimensionality was evaluated with a bi-factor model. For convergent validity, correlations with the OPD SQS, PHQ-4, and POMS were calculated. While acceptance was good, internal consistencies (ω = 0.65–0.79) and factor structure of the original four subscales were not acceptable (TLI = 0.87, CFI = 0.91, RMSEA = 0.071). EFA indicated a 3-factor solution, which was not confirmed by CFA (TLI = 0.89, CFI = 0.91, RMSEA = 0.073). Correlations between subscales and bi-factor analyses indicated an underlying general factor (TLI = 0.94, CFI = 0.96, RMSEA = 0.053). A shortened 6-item version was comparable to the original scale. Age and sex-specific representative norm-values are presented. The MZQ is a feasible, reliable and valid self-report instrument to measure representations of inner mental states. However, when applied to non-clinical samples, the total score of the MZQ should be used.

## 1. Introduction

Mentalization is a concept introduced by Fonagy, Bateman, Target and colleagues to gain a better understanding—and thereby treatment angle—for patients with borderline personality disorder (BPD), resulting in the development of a specific and effective treatment program called ‘mentalization based treatment’ (MBT) [1,2,3,4,5,6]. Ever since, mentalization has become relevant not only for BPD but also a variety of other diagnoses [7] such as depression [8,9,10], posttraumatic stress disorder [11], psychotic disorders [12,13,14], drug addiction [15], eating disorders [16,17,18], or obsessive compulsive personality disorder [19].

Mentalization or the ability to mentalize can be conceptualized as a mental process that allows for the understanding and representation of inner mental states in oneself and others by considering thoughts, needs, emotions, wishes, and desires oneself might have as well as others [7,10,20]. Assumedly, mentalization develops during the fourth and fifth year of life [20,21], is closely linked to emotion regulation [21,22], and its development is facilitated by secure attachments and relationships [16,21]. Emotional mirroring processes at infancy might play an important role in the development of mental representations of internal states, and thus, provide the basis for the infants’ emerging capacity to regulate her/his own affect. Deficits in mentalization include having no emotional awareness, not being self-reflective, or equating inner mental states with outer reality [23].

Given the young age mentalization develops, early aberrations in life such as adverse childhood experiences or disturbed attachments might be closely associated with impaired mentalization. Recent research confirms the association between adverse childhood experiences and poorer mentalization capacities [11,24], and also demonstrates the key role of mentalization as a mediator between adverse childhood experiences and adult psychopathologies like dissociation [24] or posttraumatic symptomatology [11]. The concept of mentalization might help to better understand associations between, e.g., adverse childhood experiences and mental health issues e.g., [25,26], and it might also prove to be an effective element of psychotherapeutic interventions [4,5,6].

For both clinical and research purposes, a feasible, valid and reliable instrument is necessary to assess mentalization. It was initially assessed with the ‘Reflective Self Functioning Scale’, an expert rating developed by Fonagy and colleagues [27]. Due to the need for an easily applicable self-rating instrument of mentalization, Hausberg and colleagues developed the ‘Mentalization Questionnaire’ (MZQ). A first validation of the MZQ in a sample of patients with mental disorders revealed acceptable reliability and sufficient validity to assess aspects of mentalization [7]. In order to generalize these findings, analyses of representative samples are necessary to test its performance in the general population.

The aim of our study was therefore to conduct a psychometric evaluation of the MZQ by testing several models on factorial validity and reliability based on data of a representative sample of the German population. In addition, we intended to provide representative norm values to assess mentalization capacities.

## 2. Materials and Methods

### 2.1. Sample and Setting

In cooperation with the independent demography research institute USUMA Berlin, data of a representative population sample was collected via interviews and self-report questionnaires. Between May and June 2020, 5418 households within 258 predefined regions were selected by a random route procedure. In households with multiple persons, one person was randomly selected using the Kish-Selection-Grid. Due to the COVID-19 pandemic, the interviews were conducted according to the applicable hygiene regulations (wearing face masks and keeping physical distance). Inclusion criteria were sufficient German language skills, an age ≥14, and verbal informed consent. The survey was conducted in accordance with the Declaration of Helsinki and fulfilled the ethical guidelines of the International Code of Marketing and Social Research Practice of the International Chamber of Commerce and the European Society of Opinion and Marketing Research. Ethical approval was obtained by the Ethics Committee of the Medical Faculty of the University of Leipzig (approval number: 043/20-ek).

### 2.2. Measures

#### 2.2.1. Mentalization Questionnaire (MZQ)

The original version of the MZQ was developed as a self-rated instrument to assess mentalization from a patient’s perspective by Hausberg and colleagues [7]. It consists of 15 items with responses ranging from 1 = ‘no agreement at all’ to 5 = ‘total agreement’ on a 5-point Likert scale. Analyses based on data from 434 German inpatients with mental disorders at three time points yielded four subscales with acceptable reliability and sufficient validity. Subscales comprised ’refusing self-reflection’ (4 items), ‘emotional awareness’ (4 items), ‘psychic equivalence mode’ (4 items), and ‘regulation of affect’ (3 items). Calculated sum scores for the total scale range from 15 to 75 with higher scores indicating lower mentalization capacities.

#### 2.2.2. Personality Functioning—Operationalized Psychodynamic Diagnosis Structure Questionnaire-Short Form (OPD-SQS)

The OPD-SQS is a 12-item self-report questionnaire to assess the level of personality functioning [28]. A total score and three subscales (self-perception, interpersonal contact, and relationship model) with four items each can be calculated. Responses range from 0 = ‘does not apply at all’ to 4 = ‘fully applies’. Calculated sum scores for each scale range from 0 to 16 (total sum score 0 to 48) with higher scores indicating more severe impairments of personality functioning. The OPD-SQS showed good internal consistency [29] as well as good validity and reliability [30,31]. In the present sample, the OPD-SQS showed good internal consistency (ω = 0.89) for the total scale, as well as for the subscales self-perception (ω = 0.85), while the internal consistency for the subscales contact (ω = 0.77) and relationship (ω = 0.79) was acceptable.

#### 2.2.3. Symptoms of Depression and Anxiety (PHQ-4)

The 4-item-version of the Patient Health Questionnaire (PHQ-4) measures mental distress within the last two weeks [32,33] by combining the GAD-2 (anxiety symptoms) and the PHQ-2 (depression symptoms). The sum score ranging from 0 to 12 is based on response options ranging from 0 = ‘not at all’ to 3 = ‘nearly every day’. In the present sample, the PHQ-4 showed good internal consistency (ω = 0.83).

#### 2.2.4. Profile of Mood States—POMS

The Profile of Mood States (POMS) is a widely applied instrument to assess an individual’s mood during the last 24 h in patients as well as non-clinical samples [34]. Response options on a 5 to 7-point Likert scale range from 0 = ‘not at all’ to 4 (6, respectively) = ‘very strong’. Recently, a 16-item short version of the POMS could be validated in a German representative population based sample showing good reliability [35]. In this sample, the POMS sub-scales showed good internal consistency (ω: dejection = 0.85, vigor = 0.86, fatigue = 0.89, anger = 0.89).

### 2.3. Statistical Analyses

Demographics for the sample are presented with means and standard deviations (SD). Validation of the MZQ included analysis of acceptance, reliability, and validity. Participants were excluded if more than 50% of MZQ items were missing. Acceptance was evaluated by analysis of missing or invalid items, with <5% missing values per item was interpreted as satisfactory. To evaluate the MZQs reliability, corrected item-total correlations (r < 0.3 was considered weak) were calculated for all items and internal consistencies (McDonald’s ω) for the total score and the proposed subscales of the MZQ. As a rule of thumb, McDonald’s ω was interpreted similarly to Cronbach’s α, meaning ω > 0.90 is considered excellent, while ω > 0.80 is considered good, ω > 0.70 acceptable, and ω < 0.70 questionable or poor [36].

Analysis of validity included evaluation of the factorial, convergent and discriminatory validity of the MZQ. To evaluate the factorial validity, confirmatory factor analysis (CFA) was performed for the proposed four-factor solution of the MZQ [7]. Since the internal consistency was not satisfactory for the four factor-solution, further analyses were conducted. To do so, the data set was randomly split in two samples (S1 and S2). Exploratory factor analyses (EFAs; maximum likelihood with oblimin direct rotations) were performed in S1. Scree plots and Eigenvalues (<1.0) were used to determine the ideal number of factors. To further confirm results, a parallel analysis using the syntax by O’Connor [37] was applied (n = 1000 parallel datasets; 95% percentile).

The results of the EFA were tested with a CFA in S2 to validate the proposed factorial structure. As Littles’ MCAR indicated that values were missing in random patterns in S2 (χ^2^ = 198.359, DF = 228, *p* = 0.922), data imputation was conducted with the Expectation-Maximization (EM) procedure. The use of complete datasets allowed for the application of modification indices in AMOS. Due to the high intercorrelation of the subscales, we tested unidimensional, bi-factor (three subscales and one general factor), as well as first- and second order (three subscales) models. To determine the models’ goodness of fit, Pearson’s chi-squared test (χ^2^), the comparative fit index (CFI) and root mean square error of approximation (RMSEA) with lower and higher bounds of the 95% confidence interval (CI) were calculated. To evaluate whether the empirical data fit the theoretical model, the *p*-value of Close Fit (PCLOSE) was calculated based on the RMSEA values, with values of *p* > 0.05 indicating close fit and *p* < 0.05 indicating worse than close model fit. Acceptable goodness of fit was defined as RMSEA values of <0.08 and CFI/TLI values > 0.90. Dimensionality was further evaluated by calculating the explained common variance (ECV), item explained common variance (IECV), the percentage of uncontaminated variance (PUC) and the Average Relative Parameter Bias (ARPB). ECV and PUC values > 0.70 and ARPB values < 10–15% indicate unidimensionality. Dimensionality indices were calculated using the Dueber Calculator [38]. Based on the dimensionality indices and the factor analyses, a short form of the scale was tested. From each factor that was identified by EFA, two items were included in a six-item short form. Items were chosen based on (I) loadings for the assigned factor as well as (II) factor loadings on the total score and (III) IECV loadings The IECV loading indicates how high items with load on the general factor rather than onto the assigned factor. If IECV loadings and factor loadings were contradictory, preference was given to factor loading and theoretical assignment.

Subsequently the convergent validity of the short and long form of the questionnaire was tested. To evaluate convergent validity, Pearson correlation coefficients were calculated with the OPD-SQS total score and subscales. Discriminant validity of the MZQ was tested by comparing the values with measures of psychopathology: POMS-16, GAD-2, and PHQ-4. We hypothesized that the MZQ would be positively correlated with the OPD-SQS total score and subscales, as well as with the PHQ-4 score and POMS subscales, except ‘vigor’ which was assumed to be negatively correlated to higher MZQ scores. Additionally, we assumed that the OPD-SQS would show higher correlations with the MZQ than the PHQ-4 and POMS-16, since it assesses aspects of personality functioning, which are theoretically more closely linked to mentalization than mood and psychological distress. Statistical analyses were performed with IBM SPSS (v22.0): IBM Corp, Armonk, NY, USA and SPSS AMOS (v24.0): IBM SPSS: Chicago, IL, USA. *p*-values < 0.05 (two-sided) were considered statistically significant.

## 3. Results

### 3.1. Sample Characteristics

Of the initial N = 2503 participants who completed the survey, n = 15 were excluded due to missing MZQ items. Among the remaining N = 2487, n = 1321 were women and n = 1 diverse. Average age was M = 46.0 (SD = 17.8). The majority completed secondary school (40.4%), was either married (41.9%) or single (39.7%), and worked full time (42.7%). Sociodemographic characteristics of the sample are presented in Table 1.

### 3.2. Acceptance

Overall acceptance of the MZQ items was satisfactory: only n = 90 (3.4%) of the participants did not answer one or more items, and only n = 12 (0.5%) participants left out more than one answer. The item with most missing values was item ten (‘Sometimes I only become aware of my feelings in retrospect’). The number of missing items was neither associated with age (*p* = 0.96) nor sex (*p* = 0.81).

### 3.3. Psychometric Properties and Reliability

Means, standard deviations, corrected item-total correlations and internal consistencies are presented in Table 2. Corrected item-total coefficients ranged between 0.34 and 0.64 and the internal consistency for the total scale was good (ω = 0.88). However, poor internal consistencies were found for three of the four subscales, namely ‘refusing self-reflection’ (ω = 0.65), ‘psychic equivalence mode’ (ω = 0.67), and ‘regulation of affect’ (ω = 0.69), while acceptable values were found for ‘emotional awareness’ (ω = 0.79).

### 3.4. Factorial Validity

The CFA of the proposed 4-factor solution showed low to borderline acceptable fit values with χ^2^ (84) = 1151.1 (*p* < 0.001), CMIN/df = 13.70, TLI = 0.87, CFI = 0.91, RMSEA = 0.071 (95% CI: 0.068–0.075; PCLOSE < 0.001). See also Figure 1.

Due to the questionable internal consistencies of the proposed subscales, the dataset was randomly split in two subsamples for further analyses. In subsample 1 (n = 1229) an exploratory factor analysis (EFA) was calculated. Bartlett’s test of sphericity (χ^2^ (105) = 6294.2, *p* < 0.001) was significant and the Kaiser–Meyer–Olkin measure verified the sampling adequacy for the analysis (KMO = 0.92). Eigenvalues and scree-plot indicated a three-factor solution, explaining 54.6% of the variance. Two items showed cross loadings and were assigned to the lower loading factor based on the items and factors content: item 5 (‘Most of the time it is better not to feel anything’) loaded on factor 1 (0.66) and factor 3 (0.47). Since item content of factor 3 focused on affect perception and differentiation and factor 1 mainly on cognition and communication, the item was assigned to factor 3 based on the analysis of the content of item and factor. Secondly, item 9 (‘Talking about feelings would mean that they become more and more powerful’) loaded on factor 1 (0.64) and factor 3 (0.61). Since factor 3 mainly focused on affect perception and -differentiation, the item was assigned to factor 3 based on the analysis of the content of item and factor. The content analysis for naming the extracted factors was independently conducted by two researchers, differences were resolved by consensus (reconciliation process). Based on the items content and in agreement with the theoretical background of mentalization based therapy, factor 1 was named ‘mental states in oneself regarding others—cognition and communication)’, factor 2 ‘mental states regarding oneself—affect-regulation’ and factor 3 ‘mental states regarding oneself—affect-perception and -differentiation’.

The newly developed three-factor solution was then tested with a CFA in subsample 2 (n = 1259). While fit indices slightly improved, the results were not satisfactory (χ^2^ (87) = 662.8 (*p* < 0.001), CMIN/df = 7.62, TLI = 0.89, CFI = 0.91, RMSEA = 0.073 (95% CI: 0.068–0.079; PCLOSE < 0.001). Internal consistencies were acceptable to good for factor 2 (ω = 0.80) and 3 (ω = 0.79), while values for factor 1 were questionable (ω = 0.69). The model is shown in Figure 2. A comparison of the assigned items for the 3- and 4-factor solution is shown in the Appendix A.

### 3.5. Dimensionality

Since the three subscales showed particularly high intercorrelations (r = 0.82–0.90; *p* < 0.001), the scales’ dimensionality was further evaluated. When comparing the bi-factor model (three subscales and a general factor; see Figure 3) with the correlated first-order model (three subscales), bi-factor indices clearly indicated a unidimensional structure of the MZQ (ECV = 0.787; PUC = 0.705; ARPB = 10.7%) and the item-based evaluation showed, that ten of the 15 items (66.7%) measured the general factor better than the assigned factor (IECVs > 0.85). Additionally, fit indices for the bi-factor solution were clearly superior to the three-factor model (χ^2^ (70) = 314.3 (*p* < 0.001), CMIN/df = 4.49, TLI = 0.94, CFI = 0.96, RMSEA = 0.053 (95%CI: 0.047–0.059; PCLOSE = 0.18).

Thus, we subsequently tested a shortened version of the MZQ for a general factor model. For this, the six items with highest IECV loadings as well as highest loadings for the assigned and the general factor were chosen (see Table 2). Model fit was borderline acceptable (χ^2^ (6) = 54.398 (*p* < 0.001), CMIN/df = 9.07, TLI = 0.94, CFI = 0.98, RMSEA = 0.081 (95% CI: 0.062–0.101; PCLOSE = 0.004) and the internal consistency was good (ω = 0.80).

### 3.6. Convergent Validity

Both the MZQ 15-item and MZQ 6-item version were correlated with the OPD-SQS total score and subscales to evaluate convergent validity (see also Table 3). The MZQ-15 and MZQ-6 total score had a correlation of r = 0.94 (*p* < 0.001).

As for the convergent validation, the correlation of the MZQ-6 total scale, MZQ-15 total score and subscales with the POMS-16 and PHQ-4 were substantially lower than with the SQS. While the POMS-16 subscales dejection, fatigue and anger as well as the PHQ-4 total score—indicating depression and anxiety—were significantly associated with lower mentalization, the POMS-16 subscale vigor was associated with higher mentalization. The MZQ-6 and MZQ-15 showed comparable correlations with the associated factors. For details refer to Table 4.

### 3.7. Norm Values—Normed Means for Different Age Groups and Gender

Table 5 shows a summary of mean scores and standard deviations for the MZQ and the three extracted subscales across all analyzed age groups for male and female participants. In our sample, age groups significantly differed in the overall MZQ-15 (F(5, 2487) = 11.468, *p* < 0.001; η^2^ = 0.023) and MZQ-6 scores (F(5, 2487) = 15.092, *p* < 0.001; η^2^ = 0.030). For the MZQ-15 total score no significant mean differences were found for gender (*p* = 0.12) nor gender*age (*p* = 0.09). For the MZQ-6 total score no gender differences (*p* = 0.21), but a significant gender*age group effect was observed (F(5, 2487) = 2.324, *p* = 0.041; η^2^ = 0.005).

## 4. Discussion

The aim of this study was a psychometric evaluation of the 15-item version of the MZQ [7] by testing several models on factorial validity and reliability and to provide representative norm values for the preferred model based on data of a representative sample of the German population. In contrast to Hausberg and colleagues who originally developed the 15-item version of the MZQ [7], our findings did not support the proposed 4-factor solution. Instead, we tested an alternative 3-factor solution identified in exploratory factor analysis. However, high intercorrelations between factors indicated a strong underlying general factor which was further supported by analyses of dimensionality. Based on these findings, we tested a six item short version (MZQ-6). Fit statistics of the MZQ-6 were comparable to the bi-factor version of the MZQ-15, while both reliability (internal consistency) and validity (correlations with OPD-SQS) of the MZQ-15 and MZQ-6 were comparable. Regarding their initial analyses, Hausberg and colleagues did not recommend using the MZQ subscales prior to further validation [7]. Given that our results neither support using the proposed four subscales nor the newly developed three subscales, we recommend using either the MZQ-15 total score or the MZQ-6 total score. While the MZQ-15 may allow a more nuanced assessment of mentalization, the MZQ-6 may be particularly feasible for screening purposes. Our data does not support the use of MZQ subscales in the general population.

However, it can be assumed that a representative sample of the general population presents overall good mentalization capacities, and hence, might not reflect specific impairments in mentalization in terms of statistically representable subscales. Given that impairments in mentalization are implicated in various psychological problems and disorders [39], and the fact that our results also suggest high associations with depressive and anxiety symptoms, future research will have to measure the MZQ in a clinical sample where it yet might yield robust results for the 3- or 4-factor solutions resulting in applicable subscales. Additionally, the construct validity of the MZQ-6 should be investigated by comparison with other mentalization-based questionnaires or interviews.

Consistent with other studies demonstrating that mentalization and personality functioning are interrelated constructs [40,41,42,43,44], we also found high correlations between the MZQ and the OPD questionnaire; here, higher impairments in personality functioning are associated with lower levels of mentalization. Zettl and colleagues suggest that disturbances in identity, self-direction, empathy, and intimacy are associated with lower levels of mentalization, whereas no to little impaired personality functioning is linked with higher levels of mentalization [40]. Early childhood relationship experiences are highly relevant for the development of personality functioning in terms of basic mental capacities such as adaptive coping behaviors and the formation of a stable identity [45,46,47]. Early disturbances in this developmental process—e.g., in form of childhood adversity and attachment disruptions—might result in both impaired personality functioning and low mentalization capacities [11,24,39] which, in turn, might lead to vulnerability for later mental health issues such as, e.g., depression, anxiety, dissociation, or self-harm [24,25,48,49,50,51]. Research on these associations can help us to get a better understanding of the inner workings of the mind and it can also inspire new perspectives regarding the psychopathology of mental health issues. For example, based on previous research on attachment and mentalization Fonagy, Luyten, and Allison recently proposed a developmental framework which suggested that the construct of epistemic trust (i.e., ‘the capacity to trust the relevance and generalizability of intentional communication’) might confer a vulnerability for the psychopathology underlying personality disorders [40,52,53]. A deeper knowledge on aspects like mentalization derived from studies using appropriate instruments such as the MZQ can further inform diagnosis, treatment, and future research.

There are potential limitations that should be considered regarding the study results. While overall quality of the data is high (unbiased representations of the general population), the cross-sectional study design limits the interpretation of results in terms of evaluation of test–retest reliability, sensitivity to change and predictive validity. In addition, the MZQ only allows for measuring representation of inner mental states in oneself, whereas external mentalizations in others are not properly addressed. Since at the time of the study design to our knowledge no other questionnaire adequately assessing mentalization has been validated in German, we chose to use associated scales (OPS-SQS, POMS-16 and PHQ-4) to evaluate convergent validity. In would be worthwhile however, to compare the MZQ to other mentalization-oriented questionnaires which have been developed in the meantime (e.g., the “Certainty About Mental States Questionnaire—CAMSQ” [54]) to evaluate content validity of the MZQ-15 and MZQ-6.

## 5. Conclusions

The MZQ is a feasible, reliable and valid self-report instrument to measure representations of inner mental states. However, when applied to non-clinical samples, the total score of the MZQ-15 or of the MZQ-6 screening scale should be used. Further research is necessary to examine the MZQ and its suggested subscales in clinical patient samples. Brief measures like the MZQ-6 make it possible to further research the meaning of mentalization in the context of mental health, and hence, can help to improve our understanding and knowledge of mentalization and its associations with other constructs like for example personality functioning or epistemic trust.

## Figures and Tables

**Figure 1 diagnostics-13-00135-f001:**
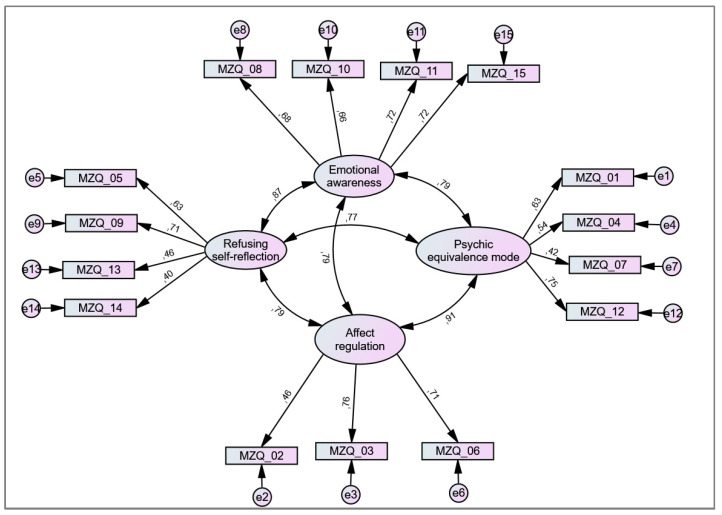
Previously proposed 4-factor solution for the Mentalization Questionnaire (MZQ). Large circles represent proposed factors, rectangles represent items (with corresponding item number), and small circles represent error terms (e). Numbers next to arrows in the model represent standardized estimates.

**Figure 2 diagnostics-13-00135-f002:**
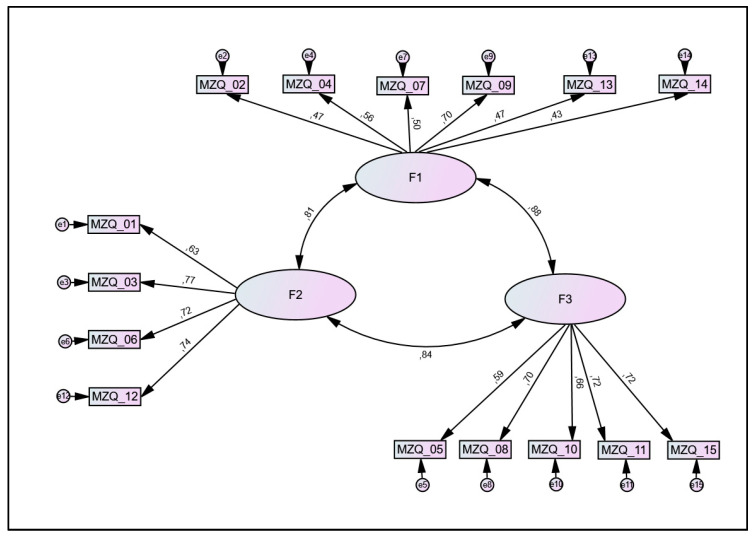
Newly introduced 3-factor solution for the Mentalization Questionnaire (MZQ). Large circles represent proposed factors, rectangles represent items (with corresponding item number), and small circles represent error terms (e). Numbers next to arrows in the model represent standardized estimates. F1 = factor 1 (‘mental states in oneself regarding others—cognition and communication’; F2 = factor 2 (‘mental states regarding oneself—affect-regulation’); F3 = factor 3 (‘mental states regarding oneself—affect-perception and -differentiation’).

**Figure 3 diagnostics-13-00135-f003:**
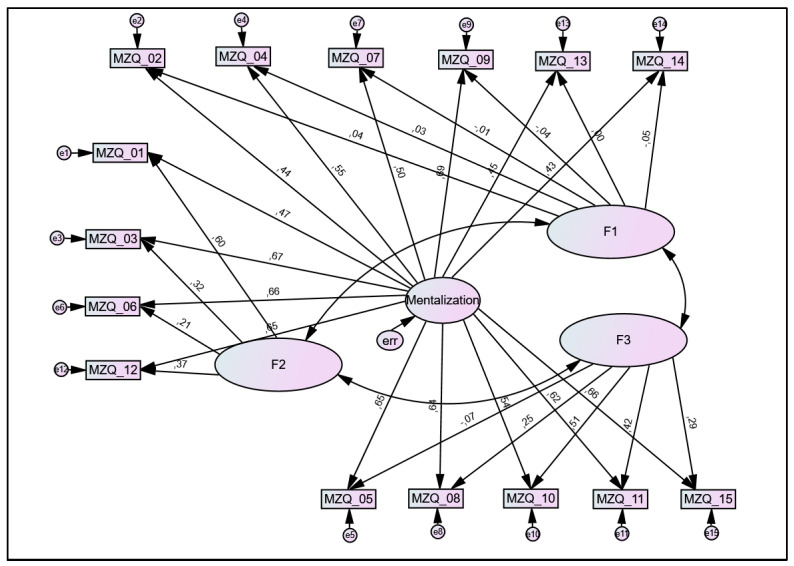
Bi-factor model for the Mentalization Questionnaire (MZQ) with a general factor and three subscales. Large circles represent proposed factors, rectangles represent items (with corresponding item number), and small circles represent error terms (e). Numbers next to arrows in the model represent standardized estimates. F1 = factor 1 (‘mental states in oneself regarding others—cognition and communication’; F2 = factor 2 (‘mental states regarding oneself—affect-regulation’); F3 = factor 3 (‘mental states regarding oneself—affect-perception and -differentiation’).

**Table 1 diagnostics-13-00135-t001:** Sociodemographic data (N = 2487).

	n	(%)
Sex		
male	1165	(46.8)
female	1321	(53.1)
diverse	1	(0.04)
Age	(M = 46.0; SD = 17.8)	
<30	613	(24.6)
30–39	358	(14.4)
40–49	369	(14.8)
50–59	527	(21.2)
60–69	381	(15.3)
≥70	239	(9.6)
Education		
no qualification	58	(2.3)
general school	505	(20.3)
secondary school	1005	(40.4)
technical college/high school	476	(19.1)
university education	360	(14.5)
other	76	(3.1)
missing	7	(0.3)
Relationship		
married	1041	(41.9)
single	988	(39.7)
divorced	293	(11.8)
widowed	149	(6.0)
missing	16	(0.6)
Employment status		
full time	1063	(42.7)
part time	407	(16.4)
unemployed	219	(8.8)
in training	249	(10.0)
retired	503	(20.2)
missing	46	(1.8)
Monthly net household income		
<1500 EUR	699	(28.1)
1500–2499 EUR	584	(23.5)
2500–3499 EUR	534	(21.5)
≥3500 EUR	670	(26.9)

**Table 2 diagnostics-13-00135-t002:** Psychometric properties and factorial structure of the MZQ and MZQ-6 short scale.

				Reliability		Factor Loading			
No	Item	M	SD	ITC	ω	EFA	CFA ^b^	CFA ^c^	IECV
	Factor 1: Mental states in oneself regarding others— cognition and communication	2.4	0.7	-	0.69				
2	Explanations from others are of little assistance in understanding my feelings.	2.6	1.1	0.44		0.43	0.47	0.45	0.96
4 ^a^	I only believe that someone really likes me a lot if I have enough realistic proof for it (e.g., a date, a gift or a hug).	2.5	1.3	0.52		0.55	0.56	0.52	0.96
7	It’s difficult for me to believe that relationships can change.	2.4	1.1	0.45		0.54	0.51	0.44	0.99
9 ^a^	Talking about feelings would mean that they become more and more powerful	2.1	1.1	0.62		0.64	0.70	0.65	1.00
13	If someone yawns in my presence, that’s a reliable sign that he is bored in my company	2.0	1.1	0.41		0.47	0.47	0.43	1.00
14	Most of the time I don’t feel like talking about my thoughts and feelings with others	2.8	1.2	0.38		0.46	0.44	0.35	0.93
	Factor 2: Mental states regarding oneself— affect-regulation	2.2	0.9		0.80				
1	If I expect to be criticized or offended, my fear increases more and more	2.5	1.2	0.51		0.74	0.58	0.55	0.49
3	Sometimes feelings are dangerous for me	2.1	1.1	0.66		0.71	0.77	0.68	0.55
6 ^a^	Often, I can’t control my feelings	2.1	1.1	0.62		0.61	0.74	0.66	0.67
12 ^a^	Often, I feel threatened by the idea that someone could criticize or offend me	2.2	1.1	0.66		0.71	0.71	0.68	0.78
	Factor 3: Mental states regarding oneself— affect-perception and -differentiation	2.3	0.8		0.79				
5	Most of the time it is better not to feel anything	2.2	1.1	0.57		0.47	0.61	0.59	1.00
8	I tend to ignore feelings of physical tension or of discomfort until they compel my full attention	2.3	1.1	0.62		0.66	0.69	0.64	0.95
10	Sometimes I only become aware of my feelings in retrospect	2.7	1.1	0.56		0.74	0.62	0.60	0.85
11 ^a^	Frequently it’s difficult for me to perceive my feelings at their full intensity	2.3	1.1	0.62		0.76	0.68	0.66	0.90
15 ^a^	Often, I don’t even know what is happening inside of me	2.0	1.0	0.65		0.68	0.72	0.70	0.48

MZQ = Mentalization Questionnaire; No = Item number; M = mean score; SD = standard deviation; ITC = item-total correlation; EFA = exploratory factor analysis; CFA = confirmatory factor analysis; ^a^ item of the 6-item version; ^b^ factor loadings for assigned factor in CFA; ^c^ factor loadings for general factor.

**Table 3 diagnostics-13-00135-t003:** Correlations of MZQ-10 and MZQ-15 total score with OPD-SQS total score and subscales.

	OPD SQS Total Score	OPD SQS Self-Perception	OPD SQS Contact	OPD SQS Relationship
MZQ-6	0.71 ***	0.69 ***	0.60 ***	0.54 ***
MZQ-15	0.73 ***	0.67 ***	0.60 ***	0.60 ***

MZQ = Mentalization Questionnaire; OPD SQS = Operationalized Psychodynamic Diagnosis Structure Questionnaire-Short Form; *** *p* < 0.001.

**Table 4 diagnostics-13-00135-t004:** Correlations of MZQ-15 and MZQ-6 total score with POMS-16 subscales scores and PHQ-4 scores.

	POMS Dejection	POMS Vigor	POMS Fatigue	POMS Anger	PHQ-4
MZQ-6 total score	0.46 ***	−0.11 ***	0.35 ***	0.36 ***	0.45 ***
MZQ-15 total score	0.46 ***	−0.22 ***	0.35 ***	0.36 ***	0.45 ***

MZQ = Mentalization Questionnaire; POMS = Profile of Mood States; *** *p* < 0.001.

**Table 5 diagnostics-13-00135-t005:** Norm values—normed sum scores for the MZQ-15 and MZQ-6 total score for male and female participants across age-groups.

			MZQ-15 Total Score		MZQ-6 Total Score	
Age	Sex	n	Mean	(SD)	Mean	(SD)
<30	M	306	34.5	(9.55)	14.0	(4.29)
	F	306	37.8	(10.39)	14.8	(5.13)
30–39	M	163	34.9	(10.69)	13.1	(4.62)
	F	196	35.6	(10.89)	13.9	(5.14)
40–49	M	181	34.5	(10.41)	12.9	(4.54)
	F	188	34.4	(10.70)	12.8	(4.81)
50–59	M	242	34.6	(9.56)	12.9	(4.24)
	F	286	33.0	(10.18)	12.1	(4.50)
60–69	M	175	31.8	(8.57)	12.0	(3.96)
	F	206	33.6	(9.94)	12.5	(4.50)
≥70	M	98	33.6	(9.64)	12.6	(4.38)
	F	140	35.2	(9.76)	13.0	(4.65)
Total sample	M	1165	34.6	(9.8)	13.1	(4.37)
	F	1322	35.0	(10.46)	13.3	(4.90)
	total	2488	34.8	(10.17)	13.2	(4.66)

M = male; F = female; n = number of participants; SD = standard deviation.

## Data Availability

The data of the study are available from the corresponding author upon reasonable request.

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
