# Peer review of "Measuring Impairments of Mentalization with the 15-Item Mentalization Questionnaire (MZQ) and Introducing the MZQ-6 Short Scale: Reliability, Validity and Norm Values Based on a Representative Sample of the German Population"

_diagnostics, 2022, doi:10.3390/diagnostics13010135_

Round 1

Reviewer 1 Report

Thank you for sending me this manuscript to review. I think it is an interesting work and very useful for the readers of this journal.

Author Response

Thank you for your kind words.

Reviewer 2 Report

I have uploaded an annotated pdf version of the originally submitted pdf file. Seventeen comments appear in the right-hand panel of the pdf file. Edits are highlighted by using the 'comments' box as well as struck-through statements. One or two citations are necessary in some sections. The authors acknowledge that the sample used in the present study is not a clinical one and this may partly account for the lower substantive convergent evidence.  

Author Response

Thank you for your helpful comments. A detailed point-to-point response is attached

Reviewer 3 Report

See the attachment.

Author Response

Thank you for your helpful comments. A detailed point-to-point response is attached. 
